# An update on the occurrence of *Paracoccidioides* species in the Midwest region, Brazil: Molecular epidemiology, clinical aspects and serological profile of patients from Mato Grosso do Sul State

**Karine Mattos**[1], **Tiago Alexandre Cocio**[2], **Edilânia Gomes Araújo Chaves**[3], **Clayton Luiz Borges**[3], **James Venturini**[1], **Lídia Raquel de Carvalho**[4], **Rinaldo Poncio Mendes**[1,5], **Anamaria Mello Miranda Paniago**[1], **Simone Schneider Weber**[6,7] *

1 Programa de Pós-Graduação em Doenças Infecciosas e Parasitárias, Faculdade de Medicina, Universidade Federal de Mato Grosso do Sul, Campo Grande, Mato Grosso do Sul, Brazil, 2 Departamento de Clínica Médica, Faculdade de Medicina de Ribeirão Preto (FMRP/USP), Universidade de São Paulo, São Paulo, Ribeirão Preto, Brazil, 3 Laboratório de Biologia Molecular, Instituto de Ciências Biológicas, Universidade Federal de Goiás, Goiânia, Goiás, Brazil, 4 Universidade Estadual Paulista (UNESP), Departamento de Bioestatística do Instituto de Biociência de Botucatu, Botucatu, São Paulo, Brazil, 5 Universidade Estadual Paulista 'Júlio Mesquita Filho' (UNESP), Faculdade de Medicina de Botucatu, São Paulo, Brazil, 6 Laboratório de Biociência (LaBio), Faculdade de Ciências Farmacêuticas, Alimentos e Nutrição, Universidade Federal de Mato Grosso do Sul, Campo Grande, Mato Grosso do Sul, Brazil, 7 Instituto de Ciências Exatas e Tecnologia (ICET), Universidade Federal do Amazonas, Itacoatiara, Amazonas, Brazil

* weberblood@gmail.com

## Abstract

### Background

Paracoccidioidomycosis (PCM) is a systemic and endemic fungal infection in Latin American, mainly in Brazil. The majority of PCM cases occur in large areas in Brazil, comprising the South, Southeast and Midwest regions, with the latter demonstrating a higher incidence of the species *Paracoccidioides lutzii*.

### Methodology and main findings

This study presents clinical, molecular and serological data of thirteen new PCM cases during 2016 to 2019 from the state of Mato Grosso do Sul, located in the Midwest region, Brazil. From these thirteen cases, sixteen clinical isolates were obtained and their genomic DNAs were subjected to genotyping by *tub1*-PCR-RFLP. Results showed *Paracoccidioides brasiliensis sensu stricto* (S1) (11/16; 68.8%), *Paracoccidioides restrepiensis* (PS3) (4/16; 25.0%) and *P. lutzii* (1/16; 6.2%) as *Paracoccidiodes* species. Therefore, in order to understand whether the type of phylogenetic species that are circulating in the state influence the reactivity profile of serological tests, we performed double agar gel immunodiffusion (DID), using exoantigens from genotyped strains found in this series of PCM cases. Overall, our DID tests have been false negative in about 30% of confirmed PCM cases. All patients were

**Data Availability Statement:** All relevant data are within the manuscript and its Supporting Information files.

**Funding:** This work was supported by Fundação de Apoio ao Desenvolvimento do Ensino, Ciência e Tecnologia do Estado de Mato Grosso do Sul (FUNDECT/DECIT-MS/CNPq/SES No. 03/2016-PPSUS-MS and PPSUS/FUNDECT No.08/2020), and Instituto Nacional de Ciência e Tecnologia (INCT) de Estratégias de Interação Patógeno-Hospedeiro (IPH) (INCT–MCTI/CNPq/CAPES/FAPs n° 16/2014). This study was financed in part by the Coordenação de Aperfeiçoamento de Pessoal de Nível Superior -Brasil (CAPES) - Finance Code 001. The doctoral grants were (KM grant numbers: 88882.458447/2019-01 and TAC grant numbers: 88882.180015/2018-01). The Universidade Federal de Mato Grosso do Sul (UFMS) supported of publication fee (EDITAL Nº 26/2021 PROP/UFMS). The funders had no role in study design, data collection and analysis, decision to publish, or preparation of the manuscript.

**Competing interests:** The authors have declared that no competing interests exist.

male, most with current or previous rural activity, with ages ranging from 17 to 59 years, with 11 patients (84.6%) over 40 years of age. No clinical or epidemiological differences were found between *Paracoccidioides* species. However, it is important to note that the only case of *P. lutzii* died as an outcome.

## Conclusions

This study suggests *P. brasiliensis sensu stricto* (S1) as the predominant species, showing its wide geographic distribution in Brazil. Furthermore, our findings revealed, for the first time, the occurrence of *P. restrepiensis* (PS3) in the state of Mato Grosso do Sul, Brazil. Despite our setbacks, it would be interesting to provide the complete sequencing of these clinical isolates to complement the molecular information presented.

## Author summary

*Paracoccidioides* spp. complex and *P. lutzii* are etiological agents of paracoccidioidomycosis (PCM), one of the most important systemic mycosis of Latin America. This study aimed to describe the molecular epidemiology associated with clinical and serological data of *Paracoccidiodes* spp. in the Mato Grosso do Sul, located in the Midwest region, Brazil. Thus, for the first time in this state, the clinical *Paracoccidioides* species were molecularly identified. Previous findings have frequently been misinterpreted as proving that *P. lutzii* predominates in this region. In fact, we observed that this state differs from others in the Midwest, presenting a higher proportion of PCM cases due to *P. brasiliensis sensu stricto* (S1). In addition, we described the first report of *P. restrepiensis* (PS3) occurrence in the Brazilian Midwest region. Then, we presented an update on the occurrence of *Paracoccidioides* species in the Midwest region from Brazil. On the other hand, our results do not demonstrate clinical and epidemiological differences between *Paracoccidioides* spp. Nevertheless, this study reinforces that other researches should be carried out in order to evaluate an association of clinical manifestations and epidemiology with *Paracoccidioides* species.

## Introduction

Paracoccidioidomycosis (PCM) is an endemic systemic fungal infection exclusive to Latin American countries such as Brazil, Argentina, Colombia and Venezuela, where approximately 10 million people have already been infected [1]. Infection in humans with *Paracoccidiodes* spp. occurs through the development of activities that involve the management of soil contaminated by conidia, such as agriculture, gardening, soil preparation and earthworks [2]. The chronic form of PCM affects mainly adults aged 30 years or older, usually male patients. The acute/sub-acute form of PCM occurs mainly in children and young adults, who represent approximately 10% to 25% of PCM cases [3].

Etiological agents of PCM are thermodimorphic fungi belonging to the *Paracoccidioides* genus, Ajellomycetaceae family, Onygenales order and Eurotiomycetes class [4]. Currently *Paracoccidioides* spp. complex is composed of five phylogenetic species: *Paracoccidiodes brasiliensis sensu stricto* (S1a and S1b), belonging to the paraphyletic group distributed in Brazil, Argentina, Paraguay, Peru and Venezuela; *Paracoccidiodes americana* (PS2), belonging to the

monophyletic group distributed in Brazil and Venezuela; *Paracoccidiodes restrepiensis* (PS3), belonging to the monophyletic group found mainly in Colombia; and *Paracoccidiodes venezuelensis* (PS4), belonging to the monophyletic group found exclusively in Venezuela [5–8]. Meanwhile, the *Paracoccidiodes lutzii* genotype includes just one species, which is the etiologic agent found in the area endemic to the states of Mato Grosso and Goiás, located in the Midwest region of Brazil, as well as in the Amazon [7,9–12].

PCM diagnosis is performed by visualization or isolation and culture of the fungus and indirectly by detection of antibodies in serological tests, where the reactivity and specificity of the tests are directly related to the preparation of exoantigens produced *in house* [13]. In addition to its low sensitivity in serological analyses, *Paracoccidioides* spp. may also show cross reactions with other microorganisms, such as *Histoplasma* spp., *Candida* spp. and *Aspergillus* spp [14]. Therefore, the observation of *Paracoccidioides* spp. suggestive structures using the microbiological (fresh examination or culture) and histopathological techniques is considered as the gold standard diagnosis [15].

Molecular biology tools have been a great ally in the identification of phylogenetic species of the genus *Paracoccidioides* spp. for genotypic studies and clinical diagnosis performed directly from samples of patients with PCM [5,11,12,16,17]. Some methodologies such as PCR–Nested [18], conventional PCR [16], qualitative PCR real time [19,20], microsatellites [8,21], mitochondrial markers [22], Multilocus Sequence Typing (MLST) [5] and whole genomic sequencing [23] have been used for the purpose of phylogenetic understanding and for answering important questions that relate genotypic data to epidemiological and serological data. A technique that identifies the species (not variety) of the *Paracoccidioides* spp. complex and *P. lutzii*, both environmental and human samples, is the Polymerase Chain Reaction—Restriction Fragment Length Polymorphism (PCR-RFLP) of the alpha tubulin (*tub*1) gene [24].

The species *P. brasiliensis sensu stricto* (S1a and S1b) has a geographic distribution in the South and Southeast regions of Brazil and its genotypic frequency is found in the states of São Paulo, Rio de Janeiro, Minas Gerais and Paraná, both in clinical and environmental samples [2,13,18,24–28]. *P. americana* (PS2) has its genotype frequency in several regions of Brazil, mainly in the South (States of Paraná and Rio Grande do Sul) and Southeast (States of São Paulo and Rio de Janeiro) [12,24,26–28]. The phylogenetic species *P. restrepiensis* (PS3) has been described as geographically restricted to Colombia and neighboring territories [5,8]. However, the occurrence of *P. restrepiensis* (PS3) in Brazil was first shown by Cocio *et al.* (2020) in a PCM endemic area from the Southeastern region (Ribeirão Preto, São Paulo, Brazil) [29]. In addition, this species also has been identified in Botucatu, São Paulo, Brazil [30]. In both municipalities, located in the central-west and northwest of the state of São Paulo respectively, *P. brasiliensis sensu stricto* (S1) is the prevalent species [26,28,30].

In short, the territorial distribution of *Paracoccidioides* species is not yet completely known [12,25,31]. In the state of Mato Grosso do Sul, there are no genetic data about the occurrence of *Paracoccidioides* spp. species. In this sense, our aim was to describe, for the first time, the occurrence of *Paracoccidioides* species using genotyping of clinical isolates, instead of serology. Furthermore, the study aim was also to determine the reactivity profile of exoantigens produced from genotyped strains of the *Paracoccidioides* spp. species found circulating in the state.

## Materials and methods

### Ethics statement

This study was approved by the Human Research Ethics Committee (CAAE: 69793917.0.0000.0021) from the Universidade Federal de Mato Grosso do Sul (UFMS). The

informed consent document was signed by all participants for research agreement. The parental consent for the participation of the child was obtained in writing, as well the child assent.

## Patients and methods

### Patients

**Location, period and design.** This study was conducted with PCM cases diagnosed between May 2016 and October 2019, from the reference center for Infectious and Parasitic Diseases at the UFMS hospital, located in the state of Mato Grosso do Sul, Brazil. Thirteen patients, from whom fungus of the *Paracoccidioides* genus was isolated, participated in the study.

**Clinical and sociodemographic analysis.** All patients with *Paracoccidioides* species isolated from their clinical sample cultures were included in this study. Clinical and sociodemographic data collected from the medical records database of PCM patients comprise: age, gender, municipality where they live, occupation, agricultural activity, HIV infection, symptoms, clinical form of PCM, severity, affected organs, therapeutic regimen, outcome and sequelae.

The affected organs were identified by clinical examination (skin, peripheral lymph nodes and pharynx), computed tomography (CT) (lungs, deep-chain lymph nodes, spleen, liver, central nervous system and adrenal glands) or by videoscopy (for larynx and intestine). The PCM clinical form of PCM was classified into acute/subacute or chronic, according to Mendes *et al.*, 2017 [3].

*Migratory history.* Information about patients' migratory history was obtained by questionnaire applied to each patient and/or the responsible family member. These collected data were: hometown, municipalities they have worked or lived in and previous occupations.

*Treatment.* PCM cases were treated with Amphotericin B, Itraconazole or Sulfamethoxazole-Trimethoprim combination (also known as Cotrimoxazole), according to the recommendations of the Brazilian guidelines on PCM [13].

## Methods

### Fungal strains

Sixteen clinical strains from 13 PCM cases, collected from different sources, including 5 skin lesion fragments, 4 lymph node aspirates, 4 oral lesion scrapings, 2 sputum and 1 tracheal aspiration were identified by classical methods of fungal identification and identified as belonging to the genus *Paracoccidioides* spp. These clinical isolates were maintained at 36˚C, in the form of yeast cells, in Fava-Netto´s medium for further genotyping and/or exoantigen production.

References strains used in this study are *Pb*18, representative of *P. brasiliensis sensu stricto* (S1b) [8]; *Pb*dog—EPM 194, considered *P. americana* (PS2) species [12]; T2—EPM 54, representative of *P. restrepiensis* (PS3) [24] and *Pb*01-like, which represents *P. lutzii* [7].

### Genomic DNA extraction from Paracoccidioides strains

The genomic DNAs isolated from *Paracoccidioides* yeast cells (reference and clinical strains) were obtained using the ZR Fungal/Bacterial extraction kit (Zymo Research, Irvine, California, United States), according to the manufacturer's protocol, and quantified using the NanoDrop 2000 (Thermo Fisher Scientifc Inc, Massachusetts, United States).

### Identification of Paracoccidioides species, phylogenetic analisys and geographic localization for this PCM case series

Clinical isolates from PCM patients were genotyped (*tub*1-PCR-RFLP) to determine their phylogenetic species, following previously described methodology by Roberto *et al.* (2016) [24],

with modification of primers as described by Hrycyk *et al.* (2018) [26]. Descriptive analysis of clinical and sociodemographic variables from PCM patients were presented according to the identified phylogenetic species.

The .jpg file generated in the photo, documentation of the *tub*1 gene fragments from clinical and reference strains submitted to the PCR—RFLP technique in this study, was used as input data for phylogenetic analysis using the PyEph 1.4. That is a software that extracts evolutionary data through of DNA fragments to generate phylogenetic trees using evolutionary methods available in this tool [32]. The method chosen for the phylogenetic analysis of the clinical and reference strains used in this study was Neighbor-Joining (NJ), where a dendrogram was constructed calculating the minimum evolution rate between them, only changing the value of genetic distance rate to 3%.

## Production of exoantigens and accomplishment of serological tests

The exoantigens were produced *in house*, according to the methodology used by Camargo *et al.* [14], from three genotyped species o *Paracoccidioides* genus, circulating in the state of Mato Grosso do Sul: *P. brasiliensis sensu stricto*–S1b (*Pb*18) [8], which was designated as *Pb*_Ag; *P. restrepiensis*—PS3 (EPM01-B_339) [14] named as *Pr*_Ag; and *P. lutzii* (clinical isolate from this study—MS2451) defined as *Pl*_Ag. To determine the reactivity profile of produced exoantigens, the double agar gel immunodiffusion technique (DID) was performed as described previously [33], against 13 PCM patients' sera, from the same cases from which the clinical strains were isolated.

## Statistical analysis

Statistical analysis was performed by Fisher's exact test for comparison of two frequencies from independent samples, and the Cochran Q test for comparison of occurrence of more than two dependent variables. Then, comparison of organs involved occurrence between *P. brasiliensis* and *P. restrepiensis* was analyzed by Fisher's exact test. While, multiple comparisons of its DID reactivity were performed using Cochran Q test. The *P. lutzii* data were not included because only one strain was described. Significance was set up at $p \leq 0.05$. The *statistical* software used was SAS—Statistical Analysis System, version 9.2.

## Results

### Sociodemographic and clinical aspects of this PCM case series

*The sociodemographic characteristics of the PCM patients included in the study are presented in* **Table 1**. All PCM cases, with isolated fungus from their culture exams, included in the study were male, with a mean age (SD) of 46 years (11.77), with most of them (84.6%, 11/13) over the age of 40. Regarding occupation, 46.1% (6/13) carry out activities with risk for infection by *Paracoccidioides* spp. (4 bricklayers and 2 rural workers), while 69.2% (9/13) are currently participating or have already participated in agricultural activities.

More than 90% of PCM cases were classified as chronic form (12/13), of which 53.8% had severe disease (7/13) and 10 (76.9%) reported fever, as shown in **S1 Table**. Twelve patients (92.3%) had at least one extra-pulmonary organ affected (**S2 Table** and **S1 Fig**). Most patients (53.8%, 7/13) were treated with cotrimoxazole, one died at admission, before treatment, and two died during the follow-up (**S1 Table**).

**Table 1.  Sociodemographic characterization of the studied PCM cases, Mato Grosso do Sul, Brazil (2016–2019).**

| PCM Cases | Sex | Age | Birthplace | Current local [1] | Occupation | Agricultural activity |
|:---:|---|---|---|---|---|---|
| 1 | Male | 42 | Deodápolis, MS | Itaquiraí, MS | Milkman | No |
| 2 | Male | 17 | Miranda, MS | Campo Grande, MS | Not work | No |
| 3 | Male | 59 | Campo Grande, MS | Sidrolândia, MS | Building Painter | Yes |
| 4 | Male | 56 | Pedro Juan Cabajero—PY | Ponta Porã, MS | Baker | Yes |
| 5 | Male | 27 | Maracaju, MS | Maracajú, MS | Physical Educator | No |
| 6 | Male | 45 | Paraná—PR | Campo Grande, MS | Bricklayer | Yes |
| 7 | Male | 43 | Glória de Dourados, MS | Campo Grande, MS | Bricklayer | Yes |
| 8 | Male | 56 | Aurora, CE | Rio Verde de Mato Grosso, MS | Farm work | Yes |
| 9 | Male | 56 | Tarumirim, MG | Campo Grande, MS | Bricklayer | Yes |
| 10 | Male | 56 | Guauraci, PR | Campo Grande, MS | Farm work | Yes |
| 11 | Male | 48 | Campo Grande, MS | Campo Grande, MS | Bricklayer | Yes |
| 12 | Male | 47 | Bataypora, MS | Campo Grande, MS | Ignored | Ignored |
| 13 | Male | 47 | Amambaí, MS | Campo Grande, MS | Picture Painter | Yes |

1 considered in this study as the geographic localization of PCM cases

## Identification of Paracoccidioides species and geographic localization for this PCM case series

The genotyping (*tub1*-PCR-RFLP) of these clinical isolates revealed the occurrence of three different *Paracoccidiodies* species: *P. brasiliensis sensu stricto* (S1) (11/16), *P. restrepiensis* (PS3) (4/16) and *P. lutzii* (1/16) (**Table 2** and **S2 Fig**).

All PCM cases described in this study (n = 13) are of patients currently living in Mato Grosso do Sul, Brazil. Nine of them (9/13; 69.2%) were infected by *P. brasiliensis sensu stricto* (S1), three (3/13; 7.7%) by *P. restrepiensis* (PS3) and just one (1/13; 7.7%) by *P. lutzii*. The

**Table 2.  Molecular identification of *Paracoccidioides* clinical strains in Mato Grosso do Sul, Brazil (2016–2019).**

| PCM Cases | Clinical sample [1] | Clinical strains | Genotyped species [2] |
|:---:|---|---|---|
| 1 | Sputum | MS1198 | *P. brasiliensis* sensu stricto (S1) |
|  | Lymph node aspirate | MS1185 | *P. brasiliensis* sensu stricto (S1) |
| 2 | Skin lesion | MS2185 | *P. brasiliensis* sensu stricto (S1) |
|  | Lymph node aspirate | MS2197 | *P. brasiliensis* sensu stricto (S1) |
| 3 | Oral lesion scraping | MS2591 | *P. brasiliensis* sensu stricto (S1) |
| 4 | Skin lesion | MS2572 | *P. brasiliensis* sensu stricto (S1) |
| 5 | Skin lesion | MS1222 | *P. brasiliensis* sensu stricto (S1) |
| 6 | Oral lesion scraping | MS2940 | *P. brasiliensis* sensu stricto (S1) |
| 7 | Skin lesion | MS1622 | *P. brasiliensis* sensu stricto (S1) |
| 8 | Oral lesion scraping | MS1162 | *P. brasiliensis* sensu stricto (S1) |
| 9 | Lymph node aspirate | MS2791 | *P. brasiliensis* sensu stricto (S1) |
| 10 | Skin lesion | MS1165 | *P. restrepiensis* (PS3) |
| 11 | Oral lesion scraping | MS190 | *P. restrepiensis* (PS3) |
|  | Sputum | MS180 | *P. restrepiensis* (PS3) |
| 12 | Lymph node aspirate | MS2343 | *P. restrepiensis* (PS3) |
| 13 | Tracheal aspirate | MS2451 | *P. lutzii* |

1 Samples from which were isolated the clinical strains by culture

2 by PCR-RFLP of *tub1* gene as described by Roberto et al (2016)

occurrence of *P. brasiliensis sensu stricto* (S1) did not differ from that of *P. restrepiensis* (PS3) (p>0.05), while the occurrence of this species was higher than that of *P. lutzii* (<0.001). Regarding the birthplaces, eight PCM patients were born in the state of Mato Grosso do Sul (Midwestern region, Brazil); two in the state of Paraná (Southern region, Brazil), one in the state of Minas Gerais (Southeast region, Brazil), one in the state of Ceará (Northeast region, Brazil) and another in Pedro Juan Cabajero, Paraguay. It is noteworthy that the migratory history survey showed that the only patient infected by *P. lutzii* lived for a period in the state of Rondônia (Western region, Brazil), despite having been born in the state of Mato Grosso do Sul.

**Fig 1** presents the phylogenetic analysis of the thirteen clinical isolates submitted to this study, using the NJ method to assess the genetic distance between them. The result shows that, thirteen samples evaluated phylogenetically, nine have evolutionary proximity with *Pb*18, thus belonging to the *P. brasiliensis sensu stricto* (S1). Three clinical isolates had genetic similarity with the reference strain EPM54—T2—*P. restrepiensis* (PS3), thus confirming their inclusion in this clade. Only one isolate (MS2451) had similarity with the reference strain *Pb*01, characterizing as *P. lutzii*. *Pb*01, MS2541 and EPM194-Pbdog were considered an external group into *Paracoccidioides* spp. species, showing genetic divergences in comparison to clinical isolates and reference strains of *P. brasiliensis sensu stricto* (S1) and *P. restrepiensis* (PS3), which that have genetic similarity between them.

## Serological profile for this PCM case series

The serum reactivity of the 13 patients varied according to the exoantigen used in the DID test (**S1 Table**). The positivity of the test is higher with *Pb*_Ag and *Pr*_Ag than with *Pl*_Ag. In addition, the serum positivity with homologous antigen was 77.8% for *Pb*_Ag and 66.7% for *Pr*_Ag (p = 0.87), and the homologous negativity was not different with these two antigens (p = 0.87).

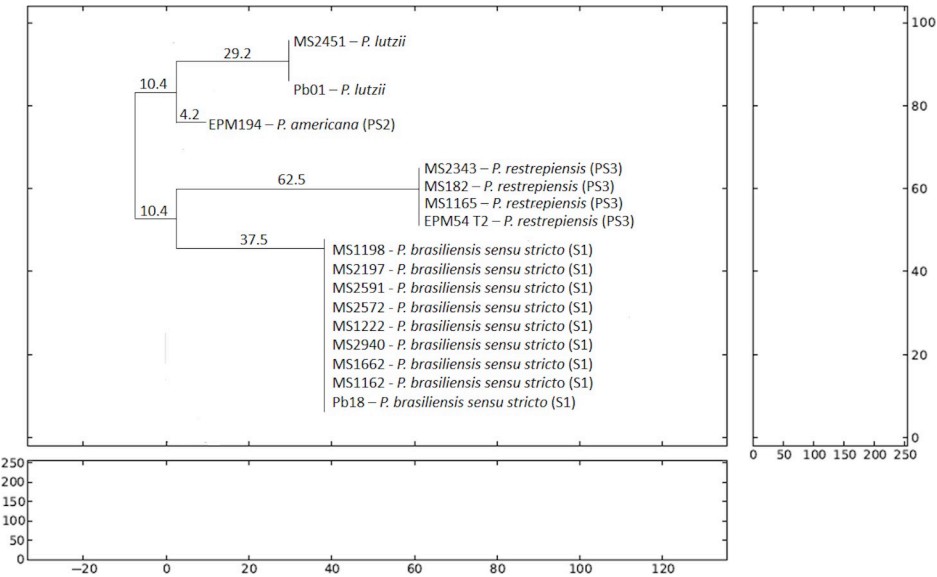

**Fig 1. Phylogenetic analysis of *tub*1 fragments generated by the PCR—RFLP technique of clinical and reference strains used in this study.** The evolutionary method used to analyze the genetic distance between them was Neighbor Joining (NJ). *Pb*01 (P. Lutzii), MS2451 (*P. Lutzii*) and EPM194 (*P. americana* (PS2)) were considered as species of close genetic variety and external group due to the phenomenon long branch attraction (LBA). *Pb*18 (*P. brasiliensis sensu stricto* (S1)) and EPM54 (*P. restrepiensis* (PS3)) had similarity with the clinical isolates classified in their respective species and showing a common similarity between them.

On the contrary, a few cross-reactions were observed with *Pl*_Ag–one with a patient infected with *P. brasiliensis sensu stricto* (S1) and one with *P. restrepiensis* (PS3) (**S1 Table**).

PCM has been confirmed in 100% of cases (13/13) by culture, and DID serology was positive in most (11/13), with 9, 9 and 3 reactive samples using *Pb*_Ag, *Pr*_Ag and *Pb*_Ag, respectively (**S1 Table**). Of these, 30.8% (4/13) of samples presented false negative results when tested with *Pb*_Ag or *Pr*_Ag, and 76.9% (10/13) presented false negative results when using *Pl*_Ag. No statistically significant differences (p≥0.05) in DID reactivity were found between S1 and PS3 cryptic species.

## Update on occurrence of Paracoccidioides species in the Midwest region

We performed a review of the literature looking for previously reports of PCM cases in the Midwest region, that have been diagnosed by molecular techniques (**S3 Table**). The geographic distribution of *Paracoccidioides* species previously reported [6,7,9,12,24,25,31,34–36] was grouped with our data to present an update on occurrence of phylogenetic species in the Midwest region, Brazil (**Fig 2**).

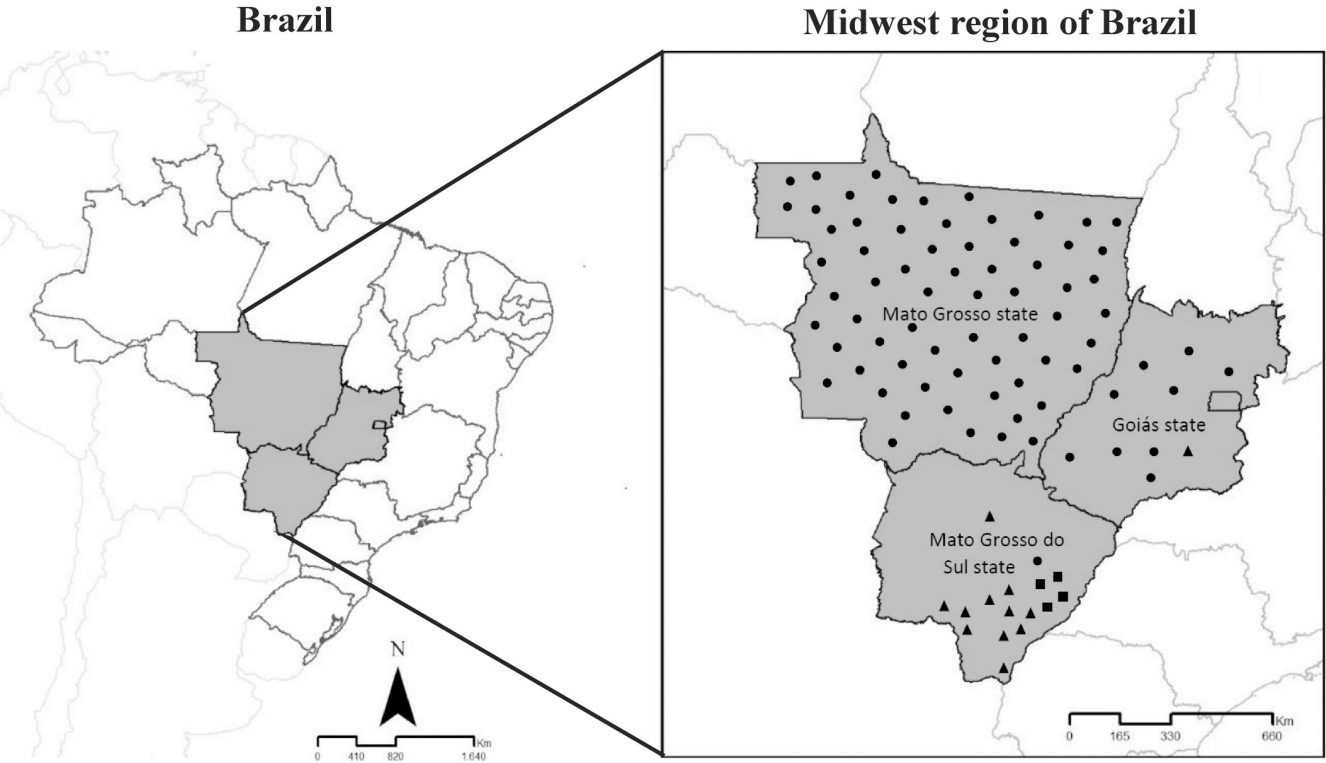

**Legend:**

- *Paracoccidioides lutzii*
▲ *Paracoccidioides brasiliensis sensu stricto* (S1)
■ *Paracoccidioides restrepiensis* (PS3)

**Fig 2. Brazil map showing an update on the *Paracoccidioides* species distribution in the Midwest region, Brazil.** The map groups all *Paracoccidioides* clinical isolates previously reported in the literature [6,7,9,12,24,25,31,33–35] for this region, additionally with found phylogenetic species from this study. The black triangles (▲) represent the geographic distribution of *P. brasiliensis sensu stricto* (S1) in Goiás and Mato Grosso do Sul (MS) states, black circles (●) indicate the presence of *P. lutzii* in the three estates, while black squares (■) are *P. restrepiensis* (PS3) found only in MS state. The map was treated using the vector graphics editor Corel Draw X8. Source: Cartographic bases (shapefiles) from Brazilian Institute of Geography and Statistics (IBGE) 2013, with maps.

## Discussion

Molecular epidemiology is revealing the geographical distribution of *Paracoccidioides* species into endemic areas [9,24,27,29,31,37]. PCM cases currently are described in almost all regions of Brazil, except the interior of the Northeast [2]. This descriptive study presents a thirteen case PCM series in the state of Mato Grosso do Sul, Brazil, from which clinical *Paracoccidioides* strains were isolated and genotyped. Thus, for the first time in this state, the clinical *Paracoccidioides* species were molecularly identified, instead of identification being based on serological data. We found nine clinical isolates genotyped as *P. brasiliensis sensu stricto* (S1), three isolates as *P. restrepiensis* (PS3), and only one as *P. lutzii*.

The best choice to identify the *Paracoccidioides* species is the molecular approach using clinical isolates from culture exam; however, it grows slowly in culture media. For example, despite this study having enrolled only 13 PCM cases, this series came from the 56 identified new cases admitted to our reference center between 2016 and 2019, and the continuous character of this procedure should be considered, in order to define the present picture and future directions.

Genotype studies evaluating clinical and environmental isolates from Brazilian South and Southeast regions, bordering the state of Mato Grosso do Sul, have shown an occurrence of *P. brasiliensis sensu stricto* (S1), confirming the predominance of this species in these areas [2,13,24–28]. In addition, *P. restrepiensis* (PS3), a phylogenetic species characterized by Matute et al. (2006), previously considered to be restricted to Colombia, was already found (two isolates—human and soil) in Venezuela in 2016, and recently in Southeastern Brazil [5,6,24,29]. Now, we present the first report of *P. restrepiensis* (PS3) occurrence in the Brazilian Midwest region. Our findings could suggest that endemic areas for PCM would be expanding from the South and Southeast regions to the Midwest and North in Brazil, maybe due to migration for the establishment of agriculture and animal husbandry [2]. However, further studies are needed to explore this potential spread.

Studies have suggested that the Brazilian Midwest region is the area in which *P. lutzii* is identified with higher frequency than in other regions, based mainly on reports from the state of Mato Grosso [7,9,12,18]. This finding was frequently misinterpreted as meaning that *P. lutzii* predominates other species in this region. In a recent publication, Hahn et al. (2019) reported that during the period 2011 to 2017 only 34 PCM patients at the reference center were treated due to *P. lutzii* in the state of Mato Grosso [9]. In addition, studies on PCM caused by *P. lutzii* from the state of Mato Grosso do Sul were performed based on serological identification [25,30]. The genotyping identification of a great number of fungi of the *Paracoccidioides* genus should be performed to identify the species distribution in different Brazilian regions.

Our findings partially differed from previous expectations, with only one *P. lutzii* isolate. The incidence rate of this disease in adults, who frequently migrate, and the long period of latency after infection, ranging from years to decades, means that it becomes difficult, if not impossible, to determine the geographic origin of these phylogenetic species. Thus, it is possible that the patients had acquired the infection in other locales. Five of 13 (38.5%) patients presented a past history of having lived in other Brazilian states. Four were born in different regions of Brazil (Northeast, South and Southeast), and currently are living in the state of Mato Grosso do Sul. For instance, one patient was born in the state of Ceará, where PCM is rare [38], and another one in Minas Gerais, an important endemic area [39,40]; both patients were infected by *P. brasiliensis sensu stricto* (S1). In addition, two patients were born in the state of Paraná—one infected by *P. restrepiensis* (PS3) and another by *P. brasiliensis sensu stricto* (S1), respectively rare and common genotypes in this Brazilian region. On the other hand, the patient infected by *P. lutzii* was born in the state of Mato Grosso do Sul, but he lived

for some time in the state of Rondônia (Northern region), where this phylogenetic species had already been reported in clinical and environmental samples [11,18].

Observing phylogenetic data of the clinical isolates by the NJ method, strains identified as *P. brasiliensis sensu stricto* (S1) and *P. restrepiensis* (PS3) presented a common ancestry, that can suggests that species 1 (*P. brasiliensis sensu stricto*—S1) suffered speciation by a possible geographic barrier generating the phylogenetic species 3 (*P. restrepiensis*—PS3) [29,31]. The strains Pb01 (*P. lutzii*), MS2451 (*P. lutzii*) and EPM194—*Pb*dog (*P. americana* (PS2)) were considered as species of the external group in this phylogenetics analisys (NJ), because they belong to species considered as high genetic divergence, respectively [5–8]. We know that the species *P. lutzii* and *P. americana* (PS2) have evolutionary distance compared to other species of the *Paracoccidioides* spp. (*P. brasiliensis sensu stricto* (S1), *P. restrepiensis* (PS3) and *P. venezuelensis* (PS4)) [3–6]. So, the phylogenetic analysis used in this study grouped these clades observing a long branch attraction (LBA) between them [41].

The evolutionary method used in our study, NJ classified *P. lutzii* and *P. americana* (PS2) as genetically close species, but it was a mistake because in the proposed nomenclature classification for species of the genus *Paracoccidioides* spp. these are considered to genetic diverge from each other [6,7]. Thus, *P. lutzii* and *P. americana* (PS2) were considered species that evolved quickly in the *Paracoccidoides* spp, phenomenon known as LBA. Due to the techniques used to identify species and the phylogenetic analysis used in this study, we are limited in terms of the availability of evolutionary methods to assess similarity between strains, but the *tub*1—PCR RFLP method and the phylogenetic tree built by the NJ method were able to answer the aspects of occurrence of *Paracoccidioides* species in the state of Mato Grosso do Sul.

Knowledge of the geographical distribution of species regarding genotype is important to define the antigen used in the serological tests, for diagnosis and control of cure, and in specific clinical, radiological and therapeutic aspects if differences are detected. Our serological results showed a high positivity for heterologous antigens between *P. brasiliensis sensu stricto* (S1) and *P. restrepiensis* (PS3), both from the *Paracoccidoides* spp. complex. Similar results were recently observed between *P. brasiliensis sensu stricto* (S1) and *P. americana* (PS2), once again between species from the *Paracoccidoides* spp. complex [27]. On the other hand, serum from only two PCM patients–one infected by *P. brasiliensis sensu stricto* (S1) and one by *P. restrepiensis* (PS3) reacted with the *P. lutzii* antigen, and serum from the patient infected by *P. lutzii* did not react with the *P. brasiliensis sensu stricto* (S1) antigen nor with the *P. restrepiensis* (PS3) antigen. This explains the previous results of false negative reactions in the serum of patients infected by *P. lutzii* tested with the antigen from *P. restrepiensis* (PS3)-B339, which belongs to the *Paracoccidioides* spp. complex. Species from the *Paracoccidioides* spp. complex present similar protein profiles, explaining the cross reactions among its cryptic species [42], while *P. lutzii* isolates present different protein profiles, which will demand future studies to define a better antigen for a more specific serological diagnosis [35,42].

The genotypic identification of clinical and environmental isolates of *Paracoccidioides* spp in countries in Latin America such as Brazil, Argentina, Paraguay and Colombia has been important to understand the geographic distribution of this species in these endemic regions of the PCM [2]. Some countries in South America have the genotypic classification determined according to the relationship between the phylogenetic frequency of each species and its geographic region. *P. restrepiensis* (PS3) was determined to be a species found exclusively in Colombia, but has been identified in countries such as Venezuela, Brazil, Argentina, Peru and Bolivia [5,23,24,29]. These reports of the presence of *P. restrepiensis* (PS3) in countries outside of Colombia suggest a possible evidence of geographic expansion of the species in countries that the presence of this clade is unexpected in other regions of South America, but genotypic studies must be carried out to respond to the genotypic frequency of species of *Paracoccidioides* spp.

genus so far exclusive in certain regions and countries of Latin America. Another species of the genus *Paracoccidioides* spp. that has been observed in a phylogenetic and genotypic study in South American countries is *P. brasiliensis sensu stricto* (S1) where, its genotypic frequency found in Argentina and Paraguay, corroborating with data from the South and Southeast of Brazil, as well as observed in the Mato Grosso do Sul, MS shown in this study [23]. *P. lutzii* was found in Ecuador [10], showing unexpected evidence of his clade outside its endemic region in the Midwest region of Brazil. With that, we observe the geographic distribution of species until then exclusive in certain countries of South America.

Our findings reinforce the importance of the identification at molecular level of the fungal species occurring in every endemic area. Which could be the key target for the best development of the diagnosis and follow-up of PCM patients.

## Supporting information

**S1 Fig. Affected organs of 13 PCM patients from the state of Mato Grosso do Sul, Brazil, with genotyping identification of the etiological agent.** Period 2016–2019. Lower-case letters compare occurrences; frequencies followed by the same letter do not differ, while those followed by different letters are statistically different (p≤0.05). Multiple comparisons were performed using Cochran Q test.
(TIF)

**S2 Fig. PCR–RFLP detection and species identification of *Paracoccidioides* genus.** Fragment standard after digestion with *Bcl*I and *Msp*I endonucleases (*tub*1 PCR-RFLP describe by Roberto et al., 2016 and modified by Hrycyk et al., 2018), showing a similarity between clinical isolates and reference strains of *Paracoccidioides* species. 3% agarose gel verifying similarity of fragments between the reference strains: **A:** *Pb*18—*P. brasiliensis sensu stricto* (S1); **B:** *Pb*dog—EPM 194—*P. americana* (PS2); **C:** EPM 54—T2—*P. restrepiensis* (PS3); **D:** *Pb*01—*P. lutzii*; and the clinical samples in this study (**1** to **13**—see in **Table 2**). **M:** 50 bp DNA ladder molecular weight marker (Sinapse Inc., USA).
(TIF)

**S1 Table. Clinical aspects and serological data of the studied PCM cases, Mato Grosso do Sul, Brazil (2016–2019).**
(XLSX)

**S2 Table. Organs involved in 13 PCM cases stratified by genotyped species.**
(XLSX)

**S3 Table. Geographic distribution of clinical isolates identified by molecular biology in the Midwest region of Brazil.**
(XLSX)

## Acknowledgments

Authors thank the Prof. Dr. Roberto Martinez from Universidade de São Paulo (USP) by support in the RFLP method and Msc. Anderson Fuentes Ferreira from Universidade Federal do Ceará (UFC) by helping in the map imagen construction.

## Author Contributions

**Conceptualization:** Clayton Luiz Borges, Rinaldo Poncio Mendes, Anamaria Mello Miranda Paniago, Simone Schneider Weber.

**Data curation:** Karine Mattos, Tiago Alexandre Cocio, Edilânia Gomes Araújo Chaves, Lídia Raquel de Carvalho, Rinaldo Poncio Mendes.

**Funding acquisition:** Clayton Luiz Borges, Anamaria Mello Miranda Paniago, Simone Schneider Weber.

**Investigation:** Karine Mattos, Clayton Luiz Borges, James Venturini, Anamaria Mello Miranda Paniago, Simone Schneider Weber.

**Software:** Simone Schneider Weber.

**Supervision:** Anamaria Mello Miranda Paniago.

**Writing – original draft:** Karine Mattos, Tiago Alexandre Cocio, Edilânia Gomes Araújo Chaves, Simone Schneider Weber.

**Writing – review & editing:** James Venturini, Lídia Raquel de Carvalho, Rinaldo Poncio Mendes, Anamaria Mello Miranda Paniago.

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
