## [Decision Letter · Decision Letter 0]

19 Nov 2020

Dear Dr Weber,

Thank you very much for submitting your manuscript "An update on  the occurrence of Paracoccidioides species in Midwest region, Brazil: molecular epidemiology, clinical aspects and serological profile of patients from Mato Grosso do Sul State." for consideration at PLOS Neglected Tropical Diseases. As with all papers reviewed by the journal, your manuscript was reviewed by members of the editorial board and by several independent reviewers. In light of the reviews (below this email), we would like to invite the resubmission of a significantly-revised version that takes into account the reviewers' comments. 

We cannot make any decision about publication until we have seen the revised manuscript and your response to the reviewers' comments. Your revised manuscript is also likely to be sent to reviewers for further evaluation.

Sincerely,

Claudio Guedes Salgado, PhD

Associate Editor

Todd Reynolds

Deputy Editor

Dear authors,

Your manuscript was reviewed by three experts in the paracoccidioidomycosis field. Although they believe the manuscript is interesting, many different queries were raised, from the way the text is organized to the necessity and quality of the figures. Please, read very carefully their concerns, answer each one of them, and change the manuscript where it is necessary.

Reviewer's Responses to Questions

**Key Review Criteria Required for Acceptance?**

**Methods**

-Are the objectives of the study clearly articulated with a clear testable hypothesis stated?

-Is the study design appropriate to address the stated objectives?

-Is the population clearly described and appropriate for the hypothesis being tested?

-Is the sample size sufficient to ensure adequate power to address the hypothesis being tested?

-Were correct statistical analysis used to support conclusions?

-Are there concerns about ethical or regulatory requirements being met?

Reviewer #1: Methods are extremely disorganized, sorted into 3 separate sections with overlapping info that should be merged and edited for clarity. Shouldn't ethics info be with patient data? the statistics portion is also a bit confusing- which data are being analyzed with which methods. The information on isolates is under patients? It seems like the diagnostics are done with in house reagents? There is very little information on the serology. that was completed- is it even necessary since you retrieved the isolates? The actual data for the strains was not presented? just the control DNA? (fig S2)

Reviewer #2: -Are the objectives of the study clearly articulated with a clear testable hypothesis stated? = YES

-Is the study design appropriate to address the stated objectives? = YES

-Is the population clearly described and appropriate for the hypothesis being tested? = YES

-Is the sample size sufficient to ensure adequate power to address the hypothesis being tested? = NO

-Were correct statistical analysis used to support conclusions? = YES

-Are there concerns about ethical or regulatory requirements being met? = YES

Reviewer #3: Are the objectives of the study clearly articulated with a clear testable hypothesis stated? objetives are learly articulated? Yes

Is the study design appropriate to address the stated objectives? Yes

Is the population clearly described and appropriate for the hypothesis being tested? Yes

Is the sample size sufficient to ensure adequate power to address the hypothesis being tested? No

Were correct statistical analysis used to support conclusions? Non applicable

Are there concerns about ethical or regulatory requirements being met? yes

 Methods are extensively described

 Description of PCM clinical forms is unnecessary in the context of your results and the low number of cases of whom 12/13 presented the chronic clinical form

 Molecular methos would be only referenced because they were already full described

 The description of the Fava Neto medium composition is enough known and can be removed.

**Results**

-Does the analysis presented match the analysis plan?

-Are the results clearly and completely presented?

-Are the figures (Tables, Images) of sufficient quality for clarity?

Reviewer #1: The story jumps around so much it is hard to follow.

Meaningless statistical analysis on site of infection.

P. lutzii missing from analysis in Fig 1 and 2?

What is the purpose of fig 3? I do not understand where these data come from

Fig S1- there are multiple indications of SNPs (missing asterisk) shown that aren't SNPs? was this alignment to include P. restrepensis?

Fig S2 would indicate that P. restrepensis does not amplify a 303bp fragment.

Reviewer #2: -Does the analysis presented match the analysis plan? = YES

-Are the results clearly and completely presented? = YES

-Are the figures (Tables, Images) of sufficient quality for clarity? Yes, But, As a request, I would like you to review figure 4, since the legend is not clear enough and it seems that a mistake was made with the symbols used: triangles (PS1), circles (PS3), squares ((PL)

Reviewer #3: Does the analysis presented match the analysis plan? Yes

Are the results clearly and completely presented? Yes

Are the figures (Tables, Images) of sufficient quality for clarity? No

There are several tables and figures which could be removed such as: Table 1; table 2 ; Table 4 ; Figure 1, Figure 2, figure 3

 This sugestion is due their repetitive information and low contribution in the context of 13 cases of PCM. Maybe one or two tables or figures can condense the data more relevant 

 It would be very interesting to present the dendogram of these species.

**Conclusions**

-Are the conclusions supported by the data presented?

-Are the limitations of analysis clearly described?

-Do the authors discuss how these data can be helpful to advance our understanding of the topic under study?

-Is public health relevance addressed?

Reviewer #1: The authors admit that only 20% of the patient samples grew isolates, but fail to recognized that the failure to grow in culture could be species specific- e.g. perhaps P. lutzii does not grow as well. 

How does understanding species in the region change treatment or diagnostic approaches. It is sort of addressed starting at line 403, but the data presented in the paper don't clearly support the claim because the numbers are not there to produce a convincing result (n=1 for P.l.). All other case data simply support previous observations.

Reviewer #2: -Are the conclusions supported by the data presented? = YES

-Are the limitations of analysis clearly described? = YES

-Do the authors discuss how these data can be helpful to advance our understanding of the topic under study? = YES

-Is public health relevance addressed? = YES

Reviewer #3: Are the conclusions supported by the data presented. More or less

Are the limitations of analysis clearly described? Yes

Do the authors discuss how these data can be helpful to advance our understanding of the topic under study? Yes

 Is public health relevance addressed? few

 I suggest that the discussion be more focussed on molecular aspects of these species than on clinical or serological data due to the low number of cases presented and most related data are enough known in the literature on PCM.

 I suggest to remark the relevance of plotting the geographical distribution of Paracoccidiodes spp in the latin American context where PCM is endemic and the potential to establish their relation with clinical forms, host preference, therapeutical outcome and balance infection/disease among others.

 Line 357: you stated that there are geographical expansion of Paracoccidiodes spp. what is the evidence about? Maybe and currently, molecular tools permit us to know how Paracoccidiodes species are distributed in areas as Mato Grosso, classically considered endemic to PCM. What do you think about?

**Editorial and Data Presentation Modifications?**

Reviewer #1: The methods section needs to be extensively revised. The authors need to clearly assess which data are new and focus on that part of the story. Much of the case data could be combined to a single table, and possibly be in supplemental.

Reviewer #2: As a request, I would like you to review figure 4, since the legend is not clear enough and it seems that a mistake was made with the symbols used: triangles (PS1), circles (PS3), squares ((PL)

Reviewer #3: Along the text there are several premises or words which could be avoided or changed such as:

Please avoid as possible words as high prevalence , higher incidence, few adequates in the PCM context

Line 39; which is one of the most instead the most important.

Line 44: accuracy instead efficacy.

Line 45 : Paracoccidiodes spp instead PCM.

Line 57: a technique that help to answer questions about epidemiological .. It is unclear. 

 Lines 66-69: But low incidence of the etiologic agent. These sentences are confuses 

 In general the text is very extensive with a lot of information already described by others along the time. I suggest it be more concise in order to catch the reader attention on molecular data

**Summary and General Comments**

Reviewer #1: The study presents information on species distribution of the causative agents of PCM in central Brazil, which represents 3 different species of Paracoccidioides. Only 20% of the patients produced cultures from samples. Some of these patients had negative serology, but the generalization of this observation is complicated by small sample size. The clinical significance is not made clear- are there differential outcomes for these patients? differing treatment regimes? There are some good points in the discussion, but the overall lack of clarity in the presentation makes it impossible to see the main, and important, conclusions.

Reviewer #2: The study aimed to describe molecular epidemiology associated with clinical and serological data of PCM in the Mato Grosso do Sul, located in the Midwest region, Brazil.

Previous findings suggested that P. lutzii predominates in this region.

The authors report a higher proportion of PCM cases due to P. brasiliensis sensu stricto (S1) and the first report of P. restrepiensis occurrence in the Brazilian Midwest region, demonstrating the geographic expansion of the genotype in South America.

Additionally, the authors show that double agar gel immunodiffusion tests have been false negative in about 30% of confirmed PCM cases.

The authors tried to demonstrate clinical and epidemiological differences between Paracoccidioides spp, but no differences could be found.

The introduction is complete and the text is well written.

The research had ethical approval.

The article is well written.

The objectives are consistent with the methodology

The methodology is explained throughout the article, clearly and progressively.

The research study period is clear.

The methodology is consistent with the objectives.

The results are very interesting and contribute to the medical literature and discussion.

Comments.

The clinical presentations of patients are too heterogeneous to allow adequate comparisons to be made.

It is very interesting how the authors took the migratory history of the patients. Since the majority of patients had chronic forms of PCM it is difficult to do a georeferencing analysis. The authors conducted a review of the literature and present a map with the updated geographical distribution of PCM species.

A statistical inference analysis of the 13 patients was performed. The sample is small and the results may have little power of statistical inference. More research is required on the geographic spread of PCM species and the clinical and epidemiological differences.

As a request, I would like you to review figure 4, since the legend is not clear enough and it seems that a mistake was made with the symbols used: triangles (PS1), circles (PS3), squares ((PL)

Reviewer #3: I carefully read your manuscript which contents novel and interesting data on Paracoccidiodes spp in Mato Grosso state of Brazil. My main concern is related with its extensive form with information already described.I remark the relevance of your results which would be more explored on the molecular and geographical context than on serological and clinical ones.In my opinion focus on molecular issues is pivotal in your paper

PLOS authors have the option to publish the peer review history of their article (what does this mean?). If published, this will include your full peer review and any attached files.

Reviewer #1: No

Reviewer #2: Yes: Deving Arias Ramos

Reviewer #3: No
---

## [Decision Letter · Decision Letter 1]

17 Mar 2021

Dear Dr Weber,

We are pleased to inform you that your manuscript 'An update on  the occurrence of Paracoccidioides species in Midwest region, Brazil: molecular epidemiology, clinical aspects and serological profile of patients from Mato Grosso do Sul State.' has been provisionally accepted for publication in PLOS Neglected Tropical Diseases.

Best regards,

Claudio Guedes Salgado, PhD

Associate Editor

Todd Reynolds

Deputy Editor

Reviewer's Responses to Questions

**Key Review Criteria Required for Acceptance?**

**Methods**

-Are the objectives of the study clearly articulated with a clear testable hypothesis stated?

-Is the study design appropriate to address the stated objectives?

-Is the population clearly described and appropriate for the hypothesis being tested?

-Is the sample size sufficient to ensure adequate power to address the hypothesis being tested?

-Were correct statistical analysis used to support conclusions?

-Are there concerns about ethical or regulatory requirements being met?

Reviewer #3: Are the objectives of the study clearly articulated with a clear testable hypothesis stated? YES

-Is the study design appropriate to address the stated objectives? YES

-Is the population clearly described and appropriate for the hypothesis being tested? YES

-Is the sample size sufficient to ensure adequate power to address the hypothesis being tested?NO

-Were correct statistical analysis used to support conclusions? YES

-Are there concerns about ethical or regulatory requirements being met?NO

**Results**

-Does the analysis presented match the analysis plan?

-Are the results clearly and completely presented?

-Are the figures (Tables, Images) of sufficient quality for clarity?

Reviewer #3: Does the analysis presented match the analysis plan? YES

-Are the results clearly and completely presented? YES

-Are the figures (Tables, Images) of sufficient quality for clarity?YES

**Conclusions**

-Are the conclusions supported by the data presented?

-Are the limitations of analysis clearly described?

-Do the authors discuss how these data can be helpful to advance our understanding of the topic under study?

-Is public health relevance addressed?

Reviewer #3: -Are the conclusions supported by the data presented? YES

-Are the limitations of analysis clearly described? YES

-Do the authors discuss how these data can be helpful to advance our understanding of the topic under study? YES

Is public health relevance addressed? YES

**Editorial and Data Presentation Modifications?**

Reviewer #3: I would like suggest to change the the present short title by "Molecular characterization of Paracoccidiodes spp in the Mato grosso do Sul, Brazil".

The Key words would be more specific such as: Paracocccidiodes brasiliensis sensu strictu(S1); Paracoccidiodes restrepiensis; Paracoccidioidomycosis: PCR RFLP.

Change the word prevalence by occurrence along the text

**Summary and General Comments**

Reviewer #3: In my opinion, the authors reviewed adequately the manuscript according with the the suggestions and comment raised during the first revision

PLOS authors have the option to publish the peer review history of their article (what does this mean?). If published, this will include your full peer review and any attached files.

Reviewer #3: No

---

## [Editor Report · Acceptance letter]

3 Apr 2021

Dear Dr Weber,

We are delighted to inform you that your manuscript, "An update on  the occurrence of Paracoccidioides species in Midwest region, Brazil: molecular epidemiology, clinical aspects and serological profile of patients from Mato Grosso do Sul State.," has been formally accepted for publication in PLOS Neglected Tropical Diseases.

Best regards,

Shaden Kamhawi

co-Editor-in-Chief

Paul Brindley

co-Editor-in-Chief
